# Beyond Osteoarthritis: Emerging Applications of Anti-NGF Monoclonal Antibodies in Pain Management in Dogs and Cats

**DOI:** 10.3390/ani15223243

**Published:** 2025-11-08

**Authors:** Giorgia della Rocca, Stefano Coaccioli, Alessandra Di Salvo

**Affiliations:** 1Department of Veterinary Medicine, University of Perugia, 06126 Perugia, Italy; alessandra.disalvo@unipg.it; 2Research Center on Animal Pain (CeRiDA), University of Perugia, 06126 Perugia, Italy; 3Italian Association for the Study of Pain (AISD), 00193 Rome, Italy; stefano.coaccioli@gmail.com

**Keywords:** mAbs anti-NGF, osteoarthritis, osteosarcoma, feline idiopathic cystitis, inflammatory bowel disease, orofacial pain, neuropathic pain, dogs, cats

## Abstract

**Simple Summary:**

Chronic pain is a major health problem that affects both humans and animals, often reducing quality of life and limiting daily activities. A key player in the development of this type of pain is a substance called Nerve Growth Factor (NGF), which normally helps nerves grow and stay healthy. However, in disease conditions, NGF can become overactive and make pain signals stronger and long-lasting. New treatments called anti-NGF monoclonal antibodies have been developed to block this process. These drugs have already shown promising results in treating painful joint disease, such as osteoarthritis, in animals like dogs and cats. Importantly, early studies suggest that their benefits may go beyond arthritis, potentially helping with other painful conditions such as certain cancers, bladder disease, gut inflammation, and nerve-related disorders. The aim of this review is to highlight how these new drugs might be used more widely in the future, what scientific evidence supports their use, and what challenges still need to be overcome. By expanding treatment options for chronic pain, these therapies could improve wellbeing for both people and animals, reducing suffering and helping patients live more comfortable lives.

**Abstract:**

Nerve Growth Factor (NGF) is a neurotrophin essential for the maintenance and growth of sensory and sympathetic nerve fibers. In pathological conditions, NGF is widely implicated in peripheral and central sensitization mechanisms, significantly contributing to the genesis and maintenance of chronic pain. Anti-NGF monoclonal antibodies, developed for the management of osteoarthritis pain, have demonstrated clinical efficacy and good tolerability in several animal species, particularly dogs and cats. Although their use is currently limited to the management of osteoarthritis pain, preclinical and clinical evidence indicates their potential therapeutic role in other pathological conditions characterized by chronic pain, such as oncological conditions like osteosarcoma, idiopathic cystitis, inflammatory bowel disease, and neuropathies. This review aims to provide an updated overview of the potential clinical applications of anti-NGF monoclonal antibodies beyond osteoarthritis, analyzing their pathophysiological rationale, available scientific evidence, possible therapeutic advantages, and limitations that remain to be addressed.

## 1. Introduction

Nerve growth factor (NGF) was initially identified as a neurotrophin essential for the survival and maintenance of the phenotype of specific subsets of peripheral sensory and sympathetic neurons as well as cholinergic nuclei of the basal forebrain during development and maturation of the nervous system. However, during adulthood, NGF is not required for survival but rather exerts a modulatory role in the physiology of sensory and nociceptive nerves, a crucial role in the generation of pain and hyperalgesia in various acute and chronic pain states. Indeed, in pathological conditions in which tissues are damaged and inflamed, NGF expression is increased. The activation of the receptor tyrosine kinase A (TrkA) by NGF on nociceptive neurons and non-neuronal cells triggers and enhances pain signaling through multiple mechanisms, contributing to peripheral and central sensitization and facilitating pain transmission [1,2]. Produced by a variety of inflammatory and immune cells (eosinophils, lymphocytes, macrophages, mast cells, Schwann cells) and following its binding to the TrkA receptor, NGF is responsible for: (i) inducing sensitization of peripheral nociceptive nerve endings through translational modifications of transient receptor potential vanilloid 1 (TRPV1), with a consequent increase in their excitability; (ii) determine an alteration in nociceptor gene transcription (transcriptional modifications), with upregulation of nociceptive receptors (TRPV1, Purinoreceptor 3 [P2X3] and Acid Sensing Ion Channel 3 [ASIC3]), neurotransmitters (Substance P, Calcitonine Gene Related Peptide [CGRP], Brain Derived Neurotrophic Factor [BDNF]), and sodium and calcium voltage-gated ion channels (Na_v_ and Ca_v_); (iii) induce the release of pro-inflammatory and pro-nociceptive mediators (e.g., histamine, prostaglandin E2, serotonin, hydrogen ions, bradykinin, and NGF itself) following its binding to TrkA receptors on mast cells; (iv) and induce nerve ending sprouting, contributing to the genesis of neuropathic pain. Through these various mechanisms, NGF causes immediate and long-term excitability, promoting the development of pain and hyperalgesia [3].

Based on this evidence, studies in rodent pain models have shown that inhibiting NGF function and signaling can block pain sensation as effectively as cyclooxygenase inhibitors and opioids. Consequently, for about two decades, several pharmaceutical companies have launched research and development programs based on a variety of approaches to antagonizing NGF, including NGF “capture,” blocking NGF binding to TrkA, and inhibiting TrkA signaling. The hypothesis was that NGF antagonism could represent a highly effective therapeutic approach, free from the adverse effects of traditional analgesics, in many painful conditions [2].

The research focused on the role of NGF in chronic osteoarthritis pain and on identifying which of the three previously proposed approaches was the most promising and effective. The first, namely the “capture” of NGF by anti-NGF monoclonal antibodies (mAbs), turned out to be the most efficacious.

## 2. Anti-NGF mAbs and Pain from Osteoarthritis

Osteoarthritis (OA) is undoubtedly one of the most common causes of chronic pain in both dogs and cats, with a prevalence ranging from 20% in dogs over one year of age to 80% in dogs over eight years of age, and 92% in cats with an average age of 9.9 years [4].

Advances in the understanding of the biology and pain mechanisms underlying OA have resulted in an increasing range of pharmaceutical treatments for canine OA over the past decade. For many years, non-steroidal anti-inflammatory drugs (NSAIDs) have been the primary option for managing pain associated with this condition. With the introduction of new drugs such as anti-NGF mAbs and piprants, along with a growing selection of adjunctive analgesics, veterinarians now have more therapeutic choices available [5]. It is also important to highlight that an effective treatment strategy for OA should be multimodal, integrating not only different pharmacological options but also approaches such as acupuncture [6], electroacupuncture [7], physiotherapy [8], nutraceuticals and phytotherapeutics [9,10], as well as other complementary therapies. This comprehensive approach is key to managing the disease as a whole [5,11].

NGF plays a key role in the pathogenesis of the disease and the associated chronic pain, exerting all the mechanisms mentioned above [12].

The first clinical studies on the potential therapeutic role of an anti-NGF mAb (tanezumab) in OA pain involved humans. However, despite high efficacy, side effects characterized by rapid progression/worsening of the arthritic condition (rapidly progressive osteoarthritis-RPOA) were observed in some of the treated subjects [13,14,15], which led the authorities responsible for the registration and marketing of new drugs to prohibit further studies. Pharmaceutical research has therefore shifted to the veterinary field, with specific reference to canine and feline species.

Regarding the former, an initial study conducted by Isola and colleagues in 2011 highlighted the presence of NGF in canine synovial fluid, as well as increased concentrations in dogs with chronic lameness compared to healthy dogs and dogs with acute lameness. The association between chronic lameness and elevated synovial concentrations led the authors of this study to suggest the involvement of NGF in OA inflammation and chronic pain [16]. Based on this evidence and the results obtained in humans, a caninized anti-NGF mAb (named NV-01) was produced, its pharmacokinetic properties were determined, and its efficacy was evaluated in an in vivo experimental model of inflammatory pain following repeated administration in dogs. The combination of stability, high affinity and potency, long half-life, and safety led to the suggestion of further development of the caninized anti-NGF mAb NV-01 as a therapeutic agent for the treatment of chronic pain in dogs [17]. Two pilot studies were therefore conducted using NV-01 in dogs spontaneously affected by OA, which demonstrated, using different outcomes (i.e., clinical metrology tools, such as the Canine Brief Pain Inventory [CBPI] and the Liverpool Osteoarthritis in Dogs Index [LOAD], as well as objective measurements such as the accelerometer), a clear scores improvement in treated subjects, highlighting a positive analgesic effect of the anti-NGF mAb in dogs affected by chronic OA pain [18,19]. In subsequent years, additional prospective, randomized, blinded, placebo-controlled clinical trials using bedinvetmab, a fully canine anti-NGF mAb, and a larger and more representative population of dogs with chronic OA pain have been conducted: all studies have demonstrated, based on owner and veterinarian assessments, excellent analgesic efficacy and tolerability following monthly administrations of the compound for several months of treatment [20,21]. A study aimed to evaluate the use of bedinvetmab for canine osteoarthritis in France, Germany, Italy, Spain, and the United Kingdom, and to provide a quantitative analysis of veterinarian satisfaction and real-world treatment patterns was recently published. This research provided evidence of the benefits of bedinvetmab use after its launch on the market in a large population of dogs in the five most populous countries of Western Europe. Compliance and satisfaction appeared high, and the use of other analgesic therapies to treat osteoarthritis pain was reduced in most cases after bedinvetmab administration [22]. A further very recent randomized, parallel-group clinical trial compared bedinvetmab and meloxicam for the management of canine osteoarthritis, demonstrating that both products were equally effective in managing osteoarthritis pain, with efficacy improving over time for both treatments. Bedinvetmab was associated with fewer adverse events [23].

Regarding specific safety and tolerability, four publications are currently available. A first study demonstrated the safety of bedinvetmab (1 mg/kg subcutaneously monthly for 6 months; multiple doses 3 and 10 times higher than recommended; with concomitant non-steroidal antinflammatory drugs [NSAID] administration for 2 weeks) in healthy adult laboratory Beagle dogs: no treatment-related alterations were observed in clinical assessments, neurological and ophthalmic examinations, joint radiological examinations, morphology or immune function, nor were any effects of short-term concomitant NSAID use. No treatment-emergent immunogenicity was observed [24]. In April 2025, a study was published to capture adverse event reports due to bedinvetmab from the Zoetis Global Pharmacovigilance database (from its initial introduction on 1 February 2021, to 30 June 2024), regardless of suspected causality or off-label use. The study found a very low rate of adverse events across over 18 million doses administered globally. Specifically, eight clinical signs were considered rare (1–10 events/10,000 treated animals-doses), with lack of efficacy having the highest rate (1.70), followed by polydipsia, ataxia, polyuria/pollakiuria, anorexia, lethargy, death, and emesis. Other clinical signs (diarrhoea, urinary incontinence; muscle weakness, convulsion, muscle tremor, tachypnoea, urinary tract infection, paresis, musculoskeletal disorder not otherwise specified, hyperactivity, increased blood urea nitrogen and lameness) were considered very rare (<1 event/10,000 treated animals-doses) [25]. However, Farrell and colleagues (2025) identified, through disproportionate analysis, a potential increase in musculoskeletal events such as ligament/tendon injuries, polyarthritis, fracture, musculoskeletal neoplasia and septic arthritis, which led the authors to suggest the need for further investigations and careful clinical monitoring of treated dogs [26]. One year earlier, the case of a dog that had developed a significant polyarthropathy after treatment with bedinvetmab had been described: having excluded other causes of arthropathy, and although a causal relationship between the polyarthropathy and bedinvetmab had not been found, the authors hypothesized that it could have been a case of RPOA following treatment [27].

Table 1 summarises the studies on the efficacy and safety of the anti-NGF mAb in OA pain in dogs.

Studies conducted in felines have paralleled those conducted in dogs. An initial preclinical study evaluated the affinity, potency, and immunoreactivity in vitro, as well as plasma pharmacokinetics, analgesic efficacy, and safety in vivo, of a felinized anti-NGF mAb (NV-02) in a kaolin-induced inflammatory pain model, achieving favorable results (high affinity, long plasma half-life, safety, and analgesic efficacy) [29]. In the same year, the results of a pilot, placebo-controlled, blinded, randomized clinical trial were published. The study evaluated the efficacy of NV-02 (single dose) for the treatment of pain and mobility impairment in cats with spontaneous OA using objective measures (accelerometer) and clinical metrology tools (Client-Specific Outcome Measures [CSOM] and Feline Musculoskeletal Pain Index [FMPI]). Data demonstrated a positive analgesic effect and improved mobility for 6 weeks after treatment [30]. In 2021, three papers using a fully feline anti-NGF mAb, frunevetmab, were published. In the work by Walters et al. (2021), its pharmacokinetics and immunoreactivity were tested in cats naturally affected by OA: overall, the administration of frunevetmab resulted in a very low incidence of immunogenicity, good subcutaneous bioavailability and no adverse effects [31]. Gruen et al. (2021) instead conducted a randomized, placebo-controlled, double-blind study [32], and a randomized, placebo-controlled, parallel-group, double-blind superiority study [33], in which the efficacy and safety of frunevetmab were evaluated in cats with spontaneous OA. The first study showed improvements in the CSOM and FMPI scores and a smaller reduction in weekly mean activity (measured by accelerometry) compared with pretreatment baseline activity with frunevetmab compared to placebo [32]. Similarly, the second study also demonstrated superior efficacy of frunevetmab compared to placebo in CSOM scores, global response, and veterinary assessments [33]. A few years later, in 2024, the results of a pilot study were published. The study aimed to use a commercially available accelerometer (Tractive GPS Tracker for Cats) to assess the activity level of cats with various forms of osteoarthritis in their natural habitat, before and after treatment with frunevetmab (using each cat as its own control), with the hypothesis that frunevetmab treatment would be associated with increased activity levels. Indeed, activity time was on average 30 min longer per day for frunevetmab compared to control [34].

That same year, a study was published investigating the effect of the anti-NGF mAbs bedinvetmab and frunevetmab on the quality of life of dogs and cats with osteoarthritis. This study used a validated web-based tool for owners to assess health-related quality of life. The study found that treatment with the specific mAb significantly improved the quality of life in both species [28].

Table 2 summarises the studies on the efficacy and safety of the anti-NGF mAb in OA pain in cats.

To conclude this section, it is worth mentioning that NGF, in addition to playing a key role in the development and maintenance of OA pain, may also be implicated in the pathogenesis of the disease: increased NGF expression has been associated with osteochondral angiogenesis [35] and increased osteoclast density (associated with subchondral bone marrow lesions) [36]. Angiogenesis and subchondral bone marrow lesions are at the origin of pain, but also of disease progression: some researchers suggest that angiogenesis promotes inflammation and tissue damage, driving the pathogenesis of OA [37]. Other researchers hypothesize that it may be possible to prevent or reduce joint disease and the resulting pain by reducing angiogenesis and nerve formation from subchondral bone in articular cartilage [35]. Further studies are needed to verify whether anti-NGF mAbs could be configured not only as analgesic drugs but also as Disease Modifying Osteoarthritis Drugs (DMOADs).

## 3. Rationale for Expanding the Use of Anti-NGF mAbs

The success achieved in treating OA pain opens the way to the possibility of using anti-NGF mAbs in other chronic pain-related diseases [38]. Numerous lines of evidence indicate that NGF is implicated in the pathogenesis of pain resulting from various conditions, such as osteosarcoma (OSA), feline idiopathic cystitis (FIC), chronic inflammatory bowel disease (IBD), and neuropathies.

In several animal models and in human clinical studies, modulation of NGF-TrkA signaling has resulted in a significant reduction in pain, suggesting the potential of these drugs in managing refractory chronic pain, even in non-orthopaedic settings [38].

Furthermore, as already seen for OA, NGF is implicated not only in the pain resulting from the aforementioned conditions, but also in their pathogenesis, hypothetically positioning anti-NGF therapy as disease-modifying.

## 4. Potential Clinical Applications

### 4.1. Osteosarcoma

Canine appendicular osteosarcoma (OSA) is a common bone tumor in dogs, causing severe pain associated with the release of inflammatory mediators and bone destruction. It is a highly aggressive tumor of mesenchymal origin, consisting of a population of rapidly proliferating malignant osteoblasts. Most affected dogs succumb to the locally destructive nature of the primary bone tumor or distant metastases involving the lung parenchyma. Although amputation of the affected limb and adjuvant systemic chemotherapy increase disease-free and pain-free intervals and overall survival times in dogs diagnosed with OSA, identifying new therapeutic targets to control the disease and associated pain is of paramount importance [39].

Numerous in vitro experimental studies have documented the mitogenic and antiapoptotic activity resulting from the activation of the NGF-TrkA signaling pathway in murine osteoblast cell lines and human sarcoma cell lines, confirming that this pathway can contribute to the proliferation and survival of canine OSA cells in local and metastatic tumor microenvironments. Indeed, in addition to their involvement in the nervous system, Trk receptors and neurotrophins appear to participate in homeostatic and reparative bone morphogenesis and play a role in pathological processes, including tumorigenesis. Further in vivo studies have confirmed the role of TrkA-dependent signaling in tumor development and progression (including metastasis), leading to the hypothesis that inhibiting the cascade of events produced by this neurotrophin could reduce tumorigenesis and consequently tumor hyperalgesia, thus improving patients’ quality of life. The use of anti-NGF antibodies could therefore represent a valid adjunct in the palliative management of cancer pain. Below are the results of some of the available studies supporting the role of NGF in tumorigenesis and the effects of inhibiting the NGF-TrkA axis.

In early in vitro studies of bone biology, transcription of Trk receptors and neurotrophin growth factors was demonstrated during the exponential growth phase of a murine osteoblast cell line (MC3T3-E1) [40,41]. Furthermore, exogenous NGF-mediated TrkA signaling induced MC3T3-E1 cells to actively secrete interleukin-6, a pro-inflammatory cytokine that promotes osteoclastogenesis and bone resorption [42]. In addition to cellular mitogenesis, in the murine osteoblast cell line MC3T3-E, TrkA activation by NGF reduces apoptotic DNA degradation in osteoblasts secondary to serum starvation or cytotoxic treatments [43]. TrkA expression and NGF-mediated signaling were then studied in human sarcoma cell lines. Analogously to what happens in osteoblasts, TrkA signaling participates in mitogenesis and cell survival, since blockade of TrkA signaling results in reduced cell proliferation and increased apoptosis in human rhabdomyosarcoma and leiomyosarcoma cell lines [44,45,46]. These in vitro studies support the role of NGF-mediated signaling through the TrkA receptor in osteoblast proliferation and differentiation and in the protection of proliferating osteoblastic cells from programmed cell death.

Subsequently, the expression of NGF and its receptor tyrosine kinase, and their correlation with clinical outcomes, was assessed by immunohistochemistry in two studies conducted on biopsy samples from human patients with appendiceal bone tumors, to determine whether these factors are involved in the transformation of osteogenic cells and the development of OSA. In both cases, the presence of neurotrophin and TrK receptors was detected, leading the authors to conclude that this growth factor can be considered a potential therapeutic target in the treatment of OSA [47,48].

In 2024, Lin and colleagues detected increased levels of NGF in OSA samples. In an experiment conducted on human OSA cell lines, the addition of NGF to the culture medium facilitated the polarization of macrophages from the M0 to the M2 phenotype (M2 macrophages, secreting immunosuppressive interleukins within the tumor environment, are associated with cancer progression and poor prognosis in several types of cancer–editor’s note). NGF also enhanced monocyte adhesion induced by vascular cell adhesion molecule-1 (VCAM-1) within the OSA microenvironment, reducing the levels of microRNA-513c-5p through specific signaling cascades (FAK and c-Src). In in vivo xenograft mouse models, NGF overexpression enhanced tumor growth. At the same time, oral administration of larotrectinib, a TrkA inhibitor, significantly antagonized NGF-promoted M2 macrophage expression and tumor progression. These findings highlight the role of NGF in tumor development and progression [49].

In the same year, Hou and colleagues (2024) studied the influence of NGF on the migration and metastasis of human OSA patients, demonstrating that NGF expression was significantly higher than that of other growth factors, and that NGF and matrix metallopeptidase-2 (MMP-2) expression levels were significantly higher in bone tissue samples from OSA patients compared to those from normal bone, and strongly correlated with tumor stage. These results led the authors to conclude that NGF is positively correlated with MMP-2 in human OSA tissue and promotes cancer cell metastasis by regulating MMP-2 expression. Furthermore, in in vivo experiments conducted on an orthotopic mouse model, NGF overexpression enhanced the effects of this neurotrophin on lung metastasis, and larotrectinib strongly inhibited the effect of NGF [50]. In another study published the following year, the same authors, starting from the assumption that angiogenesis plays a critical role in the development of OSA and its metastases, demonstrated, through an in vitro study on human OSA cells, that the NGF-TrkA axis promotes angiogenesis mediated by platelet-derived growth factor (PDGF) through the suppression of microRNA-29b-3p, and, again, that larotrectinib effectively reduces the migration and invasion capabilities of tumor cells in a dose-dependent manner [51].

Regarding the role of NGF in the development and maintenance of cancer pain, as well as the analgesic effects of anti-NGF therapy, two articles are worth mentioning. The first reports the results of a preclinical study evaluating whether the early or late administration of an NGF-sequestering antibody (mAb 911) could attenuate tumor-induced nerve sprouting, neuroma formation, and subsequent cancer pain. Using a mouse model of prostate cancer-induced bone pain, the anti-NGF was administered both early, starting on day 14 after tumor injection, when nerve sprouting had not yet occurred, and late, starting on day 35, when extensive nerve sprouting had already occurred. At 70 days post-tumor injection, it was found that, although early administration attenuated nociceptive behaviors of bone cancer more rapidly than late administration, both early and late administration significantly reduced nociceptive behaviors, nerve sprouting of sensory and sympathetic nerves, and neuroma formation. These results suggest that early or late blockade of the NGF/TrkA axis may attenuate nerve sprouting and cancer-induced pain [52]. The second article reports data from a randomized, double-blind, placebo-controlled, parallel-group, phase III study evaluating the efficacy and safety of tanezumab in human patients with cancer pain primarily due to bone metastases, undergoing background opioid therapy. Subjects were divided into two groups (stratified by tumor aggressiveness and presence/absence of concomitant anticancer treatment), treated with placebo or tanezumab every 8 weeks for 24 weeks (3 doses), respectively. Tanezumab met the primary outcome (change in daily average pain), demonstrating a greater improvement than placebo in mean daily pain intensity at the site of bone metastasis at week 8 compared to placebo. Tanezumab also demonstrated improvement compared to placebo in secondary measures of pain intensity before or after week 8. Despite improvements in pain intensity, no significant difference was observed between the treated and placebo groups in the patient’s global assessment of the impact of metastatic bone pain on daily activities or opioid use [53].

Regarding veterinary medicine, to the authors’ knowledge, only two studies have been conducted in the canine species. These studies investigated, respectively, TrKa expression and TrKa-dependent signaling in OSA cell lines and in primary tumors and their lung metastases [39], and the efficacy of bedinvetmab as a palliative treatment in primary bone tumors [54]. In the first study, in vitro and in vivo experiments were conducted to (1) evaluate TrkA mRNA and protein expression in canine OSA cell lines, (2) study the mitogenic and antiapoptotic effects of TrkA signaling in canine OSA cell lines, and (3) determine whether spontaneous canine OSA samples derived from primary tumors or lung metastatic lesions express the TrkA receptor. The results showed that canine OSA cell lines express mRNA and protein for TrkA, that blocking TrkA signaling with a protein kinase inhibitor or an NGF-neutralizing mAb induced apoptosis in the same cell lines, and that the majority of spontaneously occurring primary tumors and lung metastases of canine OSA express TrkA protein. The authors therefore concluded that given the relatively high and uniform expression of TrkA in canine OSA, as well as the induction of apoptosis in canine OSA cell lines after signaling blockade, TrkA could potentially serve as a novel therapeutic target in dogs diagnosed with OSA. In addition to its direct antitumor effects, inhibition of TrkA signaling could also be considered for the treatment of inflammatory and neuropathic pain associated with cancer-induced malignant osteolysis [39]. The second study is a case series reporting the results of treatment with bedinvetmab, in combination with zoledronate and analgesics, in six dogs with bone tumors. The authors assumed that (i) NGF binding to TrkA receptors promotes pain, osteoclastogenesis, bone resorption, and inhibits cell apoptosis, (ii) that blocking NGF can control tumor-induced pain and potentially have antitumor effects, as demonstrated by in vitro and in vivo studies, and (iii) that bedinvetmab is a canine-specific monoclonal antibody that targets the excess of NGF. Six dogs with primary bone tumors were treated with zolendronate (0.1–0.2 mg/kg every 4 weeks), bedinvetmab (0.5–1.0 mg/kg every 4 weeks), and additional analgesics (ketamine, NSAIDs, paracetamol, amantadine, and gabapentin). All treated dogs achieved clinical benefit, with a significant improvement in quality of life for a median of 118 days, and a subgroup of three dogs achieved prolonged survival compared to previously studied palliative treatments (they were still alive with an optimal quality of life 2, 9 and 11 months after diagnosis). These preliminary results suggest a benefit in including bedinvetmab in dogs unable to undergo surgery and/or radiation therapy due to financial or medical reasons [54].

### 4.2. Feline Idiopathic Cystitis

Feline idiopathic cystitis (FIC), a disease with interstitial cystitis (IC)/bladder pain syndrome (BPS) as its human counterpart, is characterized by chronic bladder inflammation and pain, altered voiding behaviors, and neurogenic inflammation.

Bladder inflammation reduces the perception threshold for innocuous or noxious stimuli applied to peripheral structures (referred hyperalgesia). Cystitis can induce transient or persistent plastic changes mediated in part by neurotrophins, such as NGF, which contribute to increased nociceptive input [55].

The ability of NGF to trigger bladder overactivity may depend on altered properties of sodium or potassium channels (or their expression) in bladder afferent fibers [56]. Interactions between the immune and nervous systems undoubtedly play a key role in the pathogenesis of this disease, which is still not fully elucidated. Evidence of immune involvement in IC/BPS stems from its high co-occurrence with known autoimmune diseases, altered cytokine profiles, and the infiltration of immune cells in patients. Cytokines can interact with the nervous system through NGF-mediated signaling, resulting in hypersensitization of nociceptors, inducing them to release substance P and creating a positive feedback loop of neuroinflammation [57].

Numerous studies in animal models or human patients have been conducted to verify both the actual expression and role of NGF-TrkA signaling during IC/BPS and the consequences of blocking this signaling pathway. It has emerged that both in murine models of visceral inflammation and in human patients affected by IC/BPS, NGF levels are significantly increased compared to healthy subjects [55,56,58,59,60,61], so much so that some authors have hypothesized the possibility of using this neurotrophin as a potential biomarker for the diagnosis of the pathology. In a study by Tonyali and coworkers (2017), a correlation was also identified between NGF levels in urine, the number of peripheral nerves in the bladder mucosa, and the severity of symptoms [60]. Conversely, a decrease in urinary NGF levels has been associated with greater pain relief and a positive response [58], and blocking NGF using an endogenous antibody or an antibody against the NGF receptor has prevented neural plasticity and bladder overactivity in experimental models of these conditions [56]. Similarly, one study also evaluated the use of tanezumab in managing IC/BPS pain, highlighting its efficacy compared to a placebo [62].

To the authors’ knowledge, only one study in felines is currently available, which identified increased NGF levels in the urothelium of cats with FIC compared to those observed in the urothelium of healthy bladders [63].

Given these findings, it seems clear that targeting the NGF-TrkA axis could help reduce visceral sensitization and improve patient comfort.

In a mouse model in which acute and chronic cystitis were induced by intravesical instillation of acrolein, administration of k252a, a nonspecific antagonist of Trk receptors, including TrkA, attenuated the referred mechanical hyperalgesia following the onset of cystitis. Furthermore, treatment with a specific NGF-neutralizing antiserum before acrolein instillation suppressed subsequent referred mechanical hyperalgesia [55].

A review published in 2022 focused on the potential therapeutic role of mAbs in patients with IC/BPS. Among the mAbs considered in the study, in addition to adalimumab and certolizumab (both anti-TNF-α antibodies, which showed opposing results in terms of statistically significant differences in all outcome measures compared to placebo), tanezumab was included, which showed both positive and negative efficacy results compared to placebo, leading the authors to conclude that monoclonal antibody therapy still needs further research before it can be proposed as a promising future therapeutic option for IC/BPS [64].

A meta-analysis of randomized controlled trials (RCTs) was subsequently conducted by Cao and colleagues (2024) to evaluate the efficacy and safety of monoclonal antibody therapies in IC/BPS. Five high-quality RCTs, including 263 patients with IC/BPS, were selected. The mAbs tested in the various studies (adalimumab, certolizumab, tanezumab, and furlanumab-also an anti-NGF mAb) were generally effective in treating IC/BPS. Patients treated with mAbs showed a higher satisfaction rate and lower ICSI (Interstitial Cystitis Symptom Index) scores. Furthermore, patients treated with mAbs experienced a reduction in pain and urinary frequency. No disparities in the incidence of complications were found between the mAb-treated and control groups. The results obtained therefore indicate that mAbs are effective and safe for the treatment of IC/BPS. However, future RCTs with larger sample sizes and long-term follow-up are needed [65].

All these results raise hopes for the potential therapeutic efficacy of frunevetmab for the treatment of feline idiopathic cystitis and related pain, but further studies in felines are obviously needed to confirm this hypothesis. It is worth noting that some empirical experiments conducted in this direction have indeed yielded promising results.

### 4.3. Inflammatory Bowel Disease

Inflammatory bowel disease (IBD) is a chronic disease common to humans and animals (dogs and cats). In humans, it can present as Crohn’s disease (CD) and ulcerative colitis (UC) [66]. Its etiology and pathogenesis are not yet fully elucidated. Defined as idiopathic, this disease is likely related to alterations in the intestinal wall involving immune and inflammatory cells. Although the presence of pain in this disease is not reported in the literature, it can easily be hypothesized, given its inflammatory nature.

Several studies have demonstrated that NGF is a factor involved in various visceral dysfunctions, including irritable bowel syndrome (IBS, an intestinal disease with symptoms overlapping with those of IBD–editor’s note) [61]. In human patients with IBD, a correlation between the expression of NGF and its receptor TrkA and increased pain, anxiety, depression, and visceral sensitivity has been observed [66]. However, in a study conducted in a mouse model in which colitis was experimentally induced, a reduction in the expression of various neurotrophins, including NGF, was detected in rat colonic smooth muscle cells [67]. Supporting these findings, immunoneutralization of NGF caused an exacerbation of experimentally induced colitis [68], suggesting that this factor, together with others, plays a regulatory role in intestinal inflammation [67]. However, opposite results have been observed in other studies in which blocking NGF through the administration of anti-NGF antibodies was effective in reducing colonic hyperalgesia in murine models of colitis [69,70], suggesting that NGF may represent a therapeutic target for pain related to this disease.

Although veterinary clinical studies are lacking, and there are some contradictions regarding the pathophysiological rationale, treatment with the available veterinary anti-NGF mAbs may represent an adjuvant option in the management of chronic visceral pain associated with IBD.

### 4.4. Orofacial Pain

One oral condition in which anti-NGF mAbs have been used empirically with some success is feline chronic gingivostomatitis, one of the most studied and debated conditions, mainly due to its complex etiopathogenesis. However, there are currently no data in the literature reporting the role of NGF in the pathogenesis of the disease and the associated pain and justifying the mechanism-oriented use of immunomodulatory therapy.

Another orofacial condition in cats, characterized by episodes of paroxysmal pain in the absence of obvious lesions, is feline orofacial pain syndrome (FOPS). The few articles published on the subject make no mention of a possible role of NGF in the pathogenesis of this condition and the associated pain. However, FOPS shares some similarities with trigeminal neuralgia and burning mouth syndrome (BMS), two pain conditions recognized in human medicine.

There is preclinical evidence that NGF contributes to inflammatory hyperalgesia in the orofacial region. However, the mechanisms underlying its hyperalgesic effect and its role in trigeminal neuropathic pain have not been fully elucidated. One study investigated (1) the ability of NGF to induce facial hyperalgesia and (2) the involvement of the TrkA receptor and mast cells in the pronociceptive effects of NGF in naive rats, as well as (3) the role of NGF in hyperalgesia in a model of trigeminal neuropathic pain. Injection of NGF into the upper lip of naive rats induced long-lasting hyperalgesia. Pretreatment with an anti-NGF antibody and TrkA receptor antagonists attenuated this phenomenon. In the murine model of trigeminal neuropathic pain, local treatment with anti-NGF antibody significantly reduced the development of NGF-induced hyperalgesia. Furthermore, increased levels of NGF were detected in the ipsilateral infraorbital nerve branch during peak hyperalgesia. These findings suggest that NGF is an important hyperalgesic mediator in the trigeminal system and may represent a potential therapeutic target for the management of orofacial pain conditions, including trigeminal neuropathic pain [71].

Regarding BMS (a chronic condition characterized by persistent oral burning in the absence of clinical or laboratory evidence of lesions), human studies have highlighted the possible involvement of NGF in the pathogenesis of this syndrome. Specifically, several studies have observed an increase in nerve fiber density in the oral mucosa of BMS patients, correlated with elevated NGF levels [72,73,74]. NGF may contribute to burning pain through peripheral and central sensitization mechanisms, amplifying the nociceptive response in the absence of noxious stimuli. Unfortunately, to the authors’ knowledge, no studies have currently evaluated the clinical consequences of blocking the NGF-TrkA axis for this condition.

Although empirical evidence in dogs and cats is still limited, chronic and idiopathic oral pain in small animals could represent a future area of interest for the application of anti-NGF antibodies. Future studies are therefore needed to clarify the role of NGF in the pathophysiology of veterinary oral pain and evaluate the therapeutic potential of anti-NGF mAbs such as bedinvetmab and frunevetmab in these contexts.

### 4.5. Neuropathic Pain and Chronic Post-Surgical Pain

Neuropathic pain can arise after injury to the peripheral and/or central nervous system. In both cases, data indicate that NGF participates in the pathophysiological mechanisms of pain initiation and maintenance. This neurotrophin is well known as an important mediator of inflammatory pain, produced by peripheral and immune cells. In neuropathic pain, persistent central and peripheral glial activation is crucial, as these activated cells release increased amounts of NGF, which sensitizes neuronal circuits, enhancing, exacerbating, and perpetuating pain and other abnormal sensations. Given its key role, NGF may serve as a therapeutic target for many types of pain, likely including neuropathic pain [75].

In 2022, Reis and colleagues published a systematic review examining the involvement of NGF in peripheral and central neuropathic pain, as well as the effects of its modulation on this pathophysiological process. From the articles reviewed, it emerged that NGF is widely considered a key factor in the development of neuropathic pain and a potential therapeutic target. Several studies have demonstrated upregulation of NGF levels in animal models of neuropathic pain, such as chronic sciatic nerve constriction, partial nerve injury, brachial plexus avulsion, partial sciatic nerve ligation, and nerve axotomy. In these models, an association between NGF overexpression and neuropathic pain was found, and the animals exhibited behavioral signs of increased skin sensitivity, including allodynia and hyperalgesia. Other studies have demonstrated the involvement of ipsilaterally produced NGF, most likely in the dorsal root ganglia (DRG), in pain following spinal nerve injury and chronic constriction. Many studies have also demonstrated the importance of NGF by modulating its levels. For example, treatment with antibodies directed against NGF after nerve injury reduced cold and mechanical hyperalgesia. Interestingly, anti-NGF treatment, in addition to reversing thermal hyperalgesia after chronic constriction injury to the sciatic nerve, can also block collateral axonal sprouting, a hallmark of neuropathic pain. Furthermore, NGF is also involved in sympathetic sprouting at the DRG level, as anti-NGF injection reduced this condition and alleviated neuropathic pain after nerve injury. These findings support the use of antibodies to modulate NGF and produce relevant analgesia in neuropathic pain [75].

Chronic postsurgical pain (CPSP) is a specific condition whose pathogenesis is likely due to nerve damage and subsequent neuropathy. Currently, CPSP is challenging to prevent and treat clinically due to a poor understanding of its pathogenetic mechanisms. Surgical damage induces the upregulation of voltage-gated sodium channels Na_v_1.7 in DRG neurons, suggesting that these channels are involved in the development of CPSP. Since NGF induces a long-term increase in neuronal hyperexcitability after damage, it has been hypothesized that this neurotrophin may cause long-term dysregulation of Na_v_1.7. This hypothesis was supported by a study conducted by Liu and colleagues (2021), who confirmed the involvement of Na_v_1.7 in NGF-induced pain and demonstrated that NGF could trigger the upregulation of Na_v_1.7 expression and thus support the development of CPSP in rats. Reversing Na_v_1.7 upregulation in DRG neurons could alleviate spinal sensitization. Attenuating Na_v_1.7 dysregulation in peripheral nociceptors by targeting the NGF-TrkA axis could therefore be a strategy to prevent the transition from acute postsurgical pain to CPSP [76].

## 5. Conclusions

Anti-NGF mAbs represent a novel therapeutic strategy for the management of chronic pain in dogs and cats with OA. Beyond OA, there are sound pathophysiological rationales for considering their use in conditions such as OSA, FIC, IBD, and neuropathic pain. Although further clinical studies are needed, their potential impact on animal quality of life is considerable and could revolutionize the approach to chronic pain in veterinary practice.

The application of anti-NGF mAbs beyond OA requires further controlled clinical trials. Current evidence is largely preclinical or derived from human medicine. Defining dosages, frequency of administration, and interactions with other analgesics will be crucial. Furthermore, cost and commercial availability may represent a barrier to widespread use of anti-NGF mAbs. However, interest in these drugs is growing, and the perspectives are promising. By expanding treatment options for chronic pain, these therapies could improve animal wellbeing, reducing suffering and helping patients live more comfortable lives.

Bedinvetmab and frunevetmab have a long half-life, allowing for monthly administration. Available safety data for both anti-NGF mAbs shows good tolerability. However, further investigations and careful clinical monitoring of treated OA dogs are suggested, as prolonged use could theoretically interfere with regenerative and injury-response mechanisms, although definitive evidence in veterinary medicine is lacking. Moreover, caution is warranted when using them in patients with active neoplastic disease, due to the potential impact on tissue remodeling and wound healing.

## Figures and Tables

**Table 1 animals-15-03243-t001:** Summary of the studies on the efficacy and tolerability of the anti-NGF mAb in OA pain in dogs.

Study Design, Dose, Treatment Duration and n. of Recruited Animals	Efficacy	Tolerability	Ref.
Randomized, blinded, placebo- and positive-controlled clinical trial to evaluate the efficacy of a single i.v. and s.c. dose of the caninized mAb anti-NGF NV-01 (0.2 mg/kg) in in 32 dogs (8 dogs/group) with experimentally induced paw inflammation. To evaluate the safety of NV-01, 3 dogs were treated i.v. with 2 mg/kg; to evaluate immunogenicity, after 8 months 2 out of 3 dogs were again injected with the same dose.	Following i.v. injection, significant differences in lameness scores were observed in NV-01 group compared to placebo group at day 1, 3, 6 and 7. Following s.c. injection, a significant reduction in lameness score respect to placebo was observed at day 7.	No adverse events were observed in the 2 weeks following the injection of NV-01. The antibody did not induce an acute neutralizing immunogenic response.	[17]
Randomized and blinded study on 9 dogs affected by chronic lameness to evaluate the efficacy of a single i.v. dose of NV-01 (0.2 mg/kg).	The CBPI PS and PI scores were significantly lower at 2 and 4 weeks after administration of NV-01 when compared to baseline. None of the enrolled dogs required rescue analgesia during the evaluation period.	According to owners’ opinions, no adverse events attributable to NV-01 administration occurred. The changes in body weight, rectal temperature, heart rate, respiratory rate, and results of CBCs and serum biochemical and electrolyte analyses were unremarkable.	[18]
Randomized, parallel group, stratified, double masked, placebo-controlled pilot study to evaluate the efficacy of a single i.v. dose of NV-01 (0.2 mg/kg) on degenerative joint disease pain in 26 dogs (13 dogs/group).	CBPI PS and PI scores significantly improved compared to baseline in the NV-01 group at 14 and 28 days after treatment, but not in the placebo group. CSOM scores showed a significant improvement compared to baseline at day 14 both in treated and in placebo group (with the degree of improvement significantly greater in the NV-01 group), but at day 28 only the treated group showed significant improvement. LOAD scores significantly improved overtime in the NV-01 group but not in the placebo group. Activity in the NV-01 group increased overtime compared to baseline. In the placebo group, no activity changes were observed. There were no changes detected overtime within groups for total pain score or index joint pain score. QoL score significantly improved overtime in the NV-01 group but not in the placebo group.	The only significant change among hematologic parameters within the NV-01 group was a decrease in packed cell volume. All other values remained within the reference range. No development of immunogenicity was detected at 28 days following administration of NV-01.	[19]
Comparative phase: Double-blind, randomized, multicentre, placebo-controlled study to evaluate the efficacy and tolerability of bedinvetmab, administered monthly (0.5–1.0 mg/kg s.c.) for 3 months in 287 dogs (bedinvetmab group = 141, placebo group = 146) with osteoarthritis. Single-armed continuation phase: dogs positively responding to bedinvetmab in phase I were treated for 6 additional months (*n* = 89).	Comparative phase: The treatment success, defined by a reduction ≥ 1 of CBPI PS score and ≥ 2 of CBPI PI score, was significantly greater in bedinvetmab group compared to placebo group at any assessment day. The VCA resulted in an overall significant improvement in bedinvetmab group compared to placebo group at all time points. Single-armed continuation phase: A total of 78 dogs finished phase II (62.8%); the remaining 11 dogs were removed from the study (10 dogs developed unrelated medical conditions and 1 dog worsened in OA clinical signs). During the continuation phase, PI and PS scores were maintained, and the overall VCA improvement plateaued.	An increase in aspartate aminotransferase and blood urea nitrogen concentrations was observed in bedinvetmab group compared to baseline and reference ranges. More dogs in bedinvetmab group than in placebo group were identified with decreased haemoglobin and packed cell volume. Adverse events occurred at similar frequencies in both groups and were considered typical for elder dogs affected by OA and/or associated with incidental comorbidities. Development of anti-drug antibodies was observed in 4 dogs: it was transient in two dogs and did not seem to neutralize bedinvetmab serum concentration or CBPI efficacy data; conversely, the other two dogs developed a persistent immunogenicity that, for one dog, cleared the CBPI efficacy along the study, and for the other one only at the beginning.	[20]
Randomized, double-blind, placebo controlled, multicentre, parallel-group study to evaluate the efficacy of bedinvetmab (0.5–1.0 mg/kg s.c.) administered once monthly for 3 months in 272 dogs (bedinvetmab group = 135, placebo group = 137) affected by osteoarthritis.	The treatment success, defined by a reduction ≥ 1 of CBPI PS score and ≥ 2 of CBPI PI score, resulted significantly greater in treated group than in placebo one from day 28 at any following control time points, but not at day 7 and 14. Mean CBPI PS and PI scores were numerically lower in bedinvetmab than in placebo group at all visits after enrolment; between-group differences for PS and PI scores were significant from day 28 and 14, respectively. There was a numerically higher proportion of bedinvetmab treated dogs with improvement in CBPI QoL at all visits versus placebo.	Haematological and biochemical values as well as urinary parameters were within ranges of reference. Adverse events occurred similarly in both groups and were considered not related to treatment. Anti-drug antibodies were observed in two dogs (one dog/group). For the dog in placebo group, it was considered to be a false-positive; for the dogs in bedinvetmab group, it appeared to be neutralizing based on a decrease in serum concentrations of bedinvetmab and total NGF but did not appear to affect efficacy results.	[21]
Quantitative retrospective online survey to evaluate the veterinarians’ satisfaction on alleviation of osteoarthritis pain in dogs following bedinvetmab treatment. Overall, 1932 patient record forms were collected from 375 veterinarians across five countries in Europe.	Mean satisfaction of veterinarians was 7.9 out of 10.0 for dogs treated up to 4 bedinvetmab doses (*n* = 1280), and 8.2 out of 10.0 for dogs treated with ≥ 5 bedinvetmab doses (*n* = 625). A reduction in the mean total number of other pharmacological treatments/dog from 1.9 to 1.3 was recorded after the start of bedinvetmab administration.		[22]
Randomised, open label, multicentre, parallel-group study to evaluate the efficacy of bedinvetmab versus meloxicam administration in dogs for the management of osteoarthritis pain. 52 dogs were treated twice (day 1 and 28) with bedinvetmab (0.5–1 mg/kg s.c.); 49 dogs were treated with meloxicam (first administration 0.2 mg/kg/day s.c., followed by 0.1 mg/kg/day orally for the following 55 days).	A significant reduction in the COI values was observed in both groups compared to baseline. In the bedinvetmab group, a greater mean reduction in COI scores was shown, but this was not statistically significant.	In the meloxicam group, the creatinine value significantly increased between days 1 and 56; all other values for haematology and serum chemistry analytes did not show significant changes.	[23]
Evaluation of dogs’ QoL during and after the treatment with bedinvetmab s.c. at 0.5–1 mg/kg every 28 days for a total of three treatments (*n* = 75).	The treatment with bedinvetmab resulted in a significant improvement in the QoL of dogs.		[28]
Three integrated laboratory studies to evaluate the safety of bedinvetmab in healthy adult Beagle dogs. Study 1: long-term safety study. Bedinvetmab (1, 3 and 10 mg/kg s.c.) administered every 28 days for a total of seven doses. Study 2: evaluation of the T-lymphocyte-dependent immune function. Bedinvetmab (1 mg/kg s.c.) administered monthly for a total of three doses. Study 3: short-term safety study with concomitant carprofen administration. A single dose of bedinvetmab (1mg/kg s.c.) was administered alone and concurrently to carprofen (4.4 mg/kg s.c. daily for 14 days); a group administered with carprofen alone was also included in the study. All groups of dogs consisted of 8 subjects (4 males and 4 females).		Bedinvetmab was well tolerated in all studies. No treatment-related effects were identified in clinical, neurological, and ophthalmic examinations or musculoskeletal evaluations, nor were any effects of short-term concomitant NSAID use. No significant changes in immune morphology or function were observed. Treatment-emergent immunogenicity was not observed.	[24]
Pharmacovigilance study aimed to analyse the most common adverse events and their frequency following bedinvetmab administration in dogs. Data were collected from the Zoetis Global Pharmacovigilance database, from launch of bedinvetmab in Europe on 1 February 2021 to 30 June 2024 (the frequency was calculated assuming one treated dog per one dose sold).		17,162 events were reported for 18,102,535 doses sold (overall rate of 9.48 events/10,000 dogs). The event most reported was lack of efficacy (with a rate of 1.70 events/10,000 dogs) followed by polydipsia, ataxia, polyuria/pollakiuria, anorexia, lethargy, death, and emesis (all with a rate <1.70 events/10,000 dogs). Other very rare (<1 event/10,000 dogs) events were diarrhoea, urinary incontinence; muscle weakness, convulsion, muscle tremor, tachypnoea, urinary tract infection, paresis, musculoskeletal disorder not otherwise specified, hyperactivity, increased blood urea nitrogen and lameness.	[25]
Case–control study to analyse the frequency of MSAERs following treatment with bedinvetmab in dogs to manage OA pain compared to those following the administration of six drugs with the same indications. Moreover, a series of 19 cases of dogs with suspected MSAE after treatment with bedinvetmab was analysed by a panel of 18 experts to evaluate the possible causal association with the administration of the drug.		MSAERs were more frequently reported (~9-times) in dogs treated with bedinvetmab compared to the combined total of dogs treated with the other drugs. A panel of experts raised a strong suspicion of a causal association between bedinvetmab and accelerated joint destruction.	[26]

CBPI: Canine Brief Pain Inventory; COI: Canine Orthopaedic Index; CSOM: Client specific outcome measure; i.v.: intravenous; LOAD: Liverpool Osteoarthritis in Dogs index; MSAERs: musculoskeletal adverse event reports; PI: pain interference; PS: pain severity; QoL: quality of life; s.c.: subcutaneous; VCA: Veterinarian Categorical Assessments.

**Table 2 animals-15-03243-t002:** Summary of studies on the efficacy and tolerability of the anti-NGF mAb in OA pain in cats.

Study Design, Dose, Treatment Duration and n. of Recruited Animals	Efficacy	Tolerability	Ref.
Two studies to evaluate the efficacy and safety of the felinized anti-NGF mAb NV-02. Study 1: blinded, placebo-controlled study evaluating the efficacy of NV-02 (2 mg/kg s.c.) in a kaolin-induced model of inflammatory lameness. The inflammation was induced 4 days after the administration of mAb or placebo (*n* = 15 cats/group). Study 2: 8 cats were treated with NV-02 at the dose of 2, 5.6, 16.8 and 28 mg/kg (2 cats/group).	Lameness scores in the NV-02 group were significantly lower compared to control group since the 2nd day from the kaolin injection up to the last day of assessment (7th day). No differences in the mean measurements of paw circumference were observed between groups.	No changes in behavior or body weight, or hematological and clinical chemistry values were observed up to 42 days following NV-02 administration.	[29]
Double-blind, placebo-controlled, randomized study to evaluate the effect on degenerative joint disease-associated pain and mobility impairment of the subcutaneous administration of NV-02 at 0.4 mg/kg (11 cats) or 0.8 mg/kg (12 cats) compared to placebo (11 cats).	The overall changes in activity, measured by an accelerometer, significantly increased in cats treated with NV-02 (the results of the 2 groups of treatment were combined) when compared to placebo group up to 6 weeks from administration. At the owner’s assessments by CSOM and FMPI, no significant differences were observed between combined treatment group and placebo group except for the CSOM scores that significantly increased in combined NV-02 treated group at 3 weeks from the treatment.	At 9 weeks from the treatment with NV-02, total protein and serum globulin concentrations were significantly higher in treated group compared to placebo group. Only in 3 cats, serum creatinine concentrations increased above the reference range. When the concentration values were evaluated within groups, no significant change was observed in treated or placebo group. All other observed events did not result associated with treatment with NV-02.	[30]
Three studies to evaluate the immunogenicity of frunevetmab in cats with OA. Study 1 (pharmacokinetic study): cats (*n* = 10) were treated i.v. and s.c. with frunevetmab (3 mg/kg) in a randomized, crossover study design with an administration interval of 28 days. Study 2 (pilot field study) and study 3 (pivotal field study): for treatment groups see the studies below [24,25].		No adverse events occurred in study 1. The results of the three studies attested for a very low risk to induce clinically relevant immunogenicity.	[31]
Multisite, randomized, placebo-controlled, double-masked study to evaluate efficacy and safety of frunevetmab in cats with degenerative joint disease. Group 1 (42 cats): first administration of frunevetmab i.v., followed by a second administration s.c. 28 days apart; Group 2 (43 cats): both treatments s.c.; Group 3 (41 cats): placebo. Frunevetmab dose ranged between 1 and 2.8 mg/kg.	A significant improvement in CSOM, FMPI and QoL scores and owner’s global assessment in frunevetmab-treated cats (the two groups were considered as combined), compared to placebo group was observed at day 42 and 56. No differences were observed between the two groups treated with frunevetmab. At the activity monitoring by accelerometry, all cats (treated groups and placebo group) showed a reduction in their activity (probably due to a falsely elevated baseline related to the placement of collars without an acclimation period); however, the placebo group accounted for a major reduction. No significant differences between groups were observed for Total Pain score and Total Joint Debility score at veterinary orthopedic examination.	Emesis, renal insufficiency, and dermatitis/eczema (this last related to collar application for the activity monitoring) were the most frequently reported adverse events in cats, with similar frequencies in frunevetmab and placebo group, except for skin disorders that occurred with higher frequency in treated cats.	[32]
Randomized, placebo-controlled, parallel-group, double-blind superiority study to evaluate the efficacy of frunevetmab, administered s.c. monthly (dose range 1.0–2.8 mg/kg) in cats with OA pain and mobility impairment and disability (frunevetmab group: 182 cats; placebo group: 93 cats).	A significant improvement in frunevetmab group compared to placebo group was observed for CSOM and owner’s global assessment at days28 and 56, while for veterinarian-assessed joint pain the improvement was significant at day 56 and 84.	Serious adverse events were considered not related to treatment; diarrhea and emesis events did not differ between groups. Skin adverse resulted significantly more frequently in frunevetmab group than in placebo one.	[33]
Cats affected by OA (n. 7) treated s.c. with 7 mg of frunevetmab (dose range: 0.7–2.3 mg/kg) to evaluate the activity level by an accelerometer. The study was conducted comparing the activity data after treatment with that before treatment (standard control: *n* = 4) and after a minimum of 30 days of washout from the treatment (inverse control: *n* = 3).	Mean minutes of cats’ activity per day resulted significantly increased in the frunevetmab group compared to control groups.		[34]
Evaluation of cats’ QoL during and after the treatment with frunevetmab (1–2.8 mg/kg s.c.) every 28 days for a total of three treatments (*n* = 56).	The treatment with frunevetmab resulted in a significant improvement in the QoL of cats.		[28]

CSOM: Client-specific outcome measure; i.v.: intravenous; FMPI: Feline musculoskeletal pain index, OA: osteoarthritis; QoL: quality of life; s.c.: subcutaneous.

## Data Availability

No new data were created or analyzed in this study. Data sharing is not applicable to this article.

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
