# Peer review of "Beyond Osteoarthritis: Emerging Applications of Anti-NGF Monoclonal Antibodies in Pain Management in Dogs and Cats"

_animals, 2025, doi:10.3390/ani15223243_

Round 1

Reviewer 1 Report

Comments and Suggestions for Authors

The review " Beyond osteoarthritis: emerging applications of anti-NGF monoclonal antibodies in pain management in dogs and cats " is extremely interesting and well written. Pain during osteoarthritis in dogs is a high-impact topic that often creates management problems for owners. The anti-NGF monoclonal antibody is a valid therapy that reduces pain, improves the quality of life in the dog and consequently in the owner and its beneficial action in conditions such as OSA, FIC, IBD and neuropathic pain makes the work even more of interest. The manuscript should be accepted for publication as it is. Indeed, to make it more complete, the authors could add, in the introduction, a breaf description of the presence of other treatments used to relieve pain during osteoatrosis. The bibliography shows that the most common treatments used to relieve clinical signs, reduce pain and inflammation include non-steroidal anti-inflammatory drugs (NSAIDs) considered the most effective however, their use is contraindicated in patients with kidney diseases, gastrointestinal or liver. For this reason, alternative approaches such as anti-NGF monoclonal antibodies or the administration of nutraceuticals, cannabinoids or even non-invasive treatments such as acupuncture, laser therapy or laser acupuncture can be useful in the management of OA.

Author Response

The authors warmly thank the referee for his/her appreciation of the review. Responses to the comments are provided below.

.....Indeed, to make it more complete, the authors could add, in the introduction, a breaf description of the presence of other treatments used to relieve pain during osteoatrosis. The bibliography shows that the most common treatments used to relieve clinical signs, reduce pain and inflammation include non-steroidal anti-inflammatory drugs (NSAIDs) considered the most effective however, their use is contraindicated in patients with kidney diseases, gastrointestinal or liver. For this reason, alternative approaches such as anti-NGF monoclonal antibodies or the administration of nutraceuticals, cannabinoids or even non-invasive treatments such as acupuncture, laser therapy or laser acupuncture can be useful in the management of OA.

A paragraph has been added at the beginning of the second paragraph (Anti-NGF mAbs and pain from osteoarthritis), and the related citations have been added in the references section. 

Reviewer 2 Report

Comments and Suggestions for Authors

Dear authors

The current review aims to present the current knowledge of anti-NGF monoclonal antibodies, mainly used for OA in dogs and cats, and to highlight how these agents can be used more widely in the future in other pathological situations contributing to chronic pain. In my opinion this review is very interesting, providing further perspectives on the use of monoclonal antibodies. It is well-structured and can be a good starting point for the potential incorporation of such pharmacotherapies into a unimodal or even a multimodal approach of chronic pain conditions.

I only have a few suggestions to improve the manuscript;

Introduction

Line 68: there is no term "pro-pain mediator". Please use a better term such as proniciceptive or antinociceptive mediator.

Line 92: please change "in the previous paragraph" with ''above".

I wish you good luck with your manuscript, hoping that it will contribute to the initiation of new research on such chronic pain conditions.

Author Response

The authors warmly thank the referee for his/her appreciation of the review. Responses to the comments are provided below.

Line 68: there is no term "pro-pain mediator". Please use a better term such as proniciceptive or antinociceptive mediator.

The word has been changed as suggested.

Line 92: please change "in the previous paragraph" with ''above".

The word has been changed as suggested.

Reviewer 3 Report

Comments and Suggestions for Authors

I wish to congratulate the authors for their excellent work.

The manuscript presents a comprehensive and well-structured review on the role of Nerve Growth Factor (NGF) and the therapeutic potential of anti-NGF monoclonal antibodies in the management of chronic pain in animals. The topic is of significant scientific and clinical relevance, addressing an area of growing interest in both veterinary and translational medicine. The title is appropriate and accurately reflects the content and scope of the review. The manuscript is well organized, with a logical progression from the physiological and pathophysiological background to preclinical and clinical applications. The introduction provides an adequate overview of the biological role of NGF, while the discussion effectively integrates experimental evidence and clinical findings. The conclusions are balanced and appropriately highlight both the current achievements and the challenges that remain to be addressed.

The tables are complete, clearly presented, and effectively summarize the key studies included in the review. They provide a useful and comprehensive view of the selected works, enhancing the clarity and accessibility of the information presented.

The review is written in a clear and fluent style. The scientific content is accurate, and the discussion reflects a critical and up-to-date understanding of the field. The authors successfully highlight the translational relevance of anti-NGF therapies beyond osteoarthritis, emphasizing their potential applications in a variety of chronic pain conditions.

I found the paper very enjoyable to read, and I do not consider any revisions necessary prior to publication.

Author Response

The authors warmly thank the referee for his/her appreciation of the review!